# Predictive and Prognostic Factors in Melanoma Central Nervous System Metastases—A Cohort Study

**DOI:** 10.3390/cancers16122272

**Published:** 2024-06-19

**Authors:** Estefania Serra, Álvaro Abarzua-Araya, Ana Arance, Roberto Martin-Huertas, Francisco Aya, María Lourdes Olondo, Daniel Rizo-Potau, Josep Malvehy, Susana Puig, Cristina Carrera, Sebastian Podlipnik

**Affiliations:** 1Dermatology Department, Hospital Clinic of Barcelona, University of Barcelona, 08036 Barcelona, Spain; nia720@hotmail.com (E.S.); alvaroabarzuaaraya@gmail.com (Á.A.-A.); danielrizopotau@msn.com (D.R.-P.); jmalvehy@gmail.com (J.M.); susipuig@gmail.com (S.P.); podlipnik@clinic.cat (S.P.); 2Melanoma and Skin Cancer Unit, Dermatology Department, Escuela de Medicina, Pontificia Universidad Católica de Chile, Santiago 8320165, Chile; 3Oncology Department, Hospital Clinic of Barcelona, 08036 Barcelona, Spain; amarance@clinic.cat (A.A.); rmartinh@clinic.cat (R.M.-H.);; 4Radiology Service, Hospital Clinic of Barcelona, 08036 Barcelona, Spain; mlolondo@clinic.cat; 5Institut d’Investigacions Biomediques August Pi I Sunyer (IDIBAPS), 08036 Barcelona, Spain; 6CIBER on Rare Disease, Instituto de Salud Carlos III, 28029 Barcelona, Spain

**Keywords:** melanoma, brain metastases, risk factors, microscopic satellitosis, prognostic factors, cutaneous melanoma, survival analysis

## Abstract

**Simple Summary:**

We conducted a study at the Melanoma Unit of the Hospital Clinic of Barcelona to investigate brain metastases in patients with cutaneous melanoma. We collected data from patients diagnosed between January 1998 and September 2023. Patients with melanoma in situ or those with prior lung or breast cancer were excluded. Our aim was to identify factors associated with the development and survival outcomes of brain metastases. We analyzed patient demographics, tumor characteristics, and survival data. The diagnosis of brain metastases was confirmed using imaging techniques, and biopsies were performed when feasible. Our study followed strict guidelines for reporting observational studies. We found that younger age and larger primary tumor thickness increased the risk of developing brain metastases. Additionally, the presence of ulceration and microscopic satellitosis in the primary tumor were associated with a higher risk. Melanomas located on the trunk had a higher risk compared to those on the extremities. Patients with brain metastases had a median survival of around six months. Neurological symptoms and leptomeningeal involvement were associated with poorer survival outcomes. Higher number of brain lesions and elevated levels of lactate dehydrogenase (LDH) also predicted worse survival. Our findings highlight the importance of early detection and monitoring of melanoma patients, especially those at higher risk of brain metastases. Understanding these factors can aid in personalized treatment approaches and improving patient outcomes.

**Abstract:**

Background: Melanoma is the cancer with the highest risk of dissemination to the central nervous system (CNS), one of the leading causes of mortality from this cancer. Objective: To identify patients at higher risk of developing CNS metastases and to evaluate associated prognostic factors. Methods: A cohort study (1998–2023) assessed patients who developed CNS melanoma metastases. Multivariate logistic regression was used to identify predictive factors at melanoma diagnosis for CNS metastasis. Cox regression analysis evaluated the CNS-independent metastasis-related variables impacting survival. Results: Out of 4718 patients, 380 (8.05%) developed CNS metastases. Multivariate logistic regression showed that a higher Breslow index, mitotic rate ≥ 1 mm^2^, ulceration, and microscopic satellitosis were significant risk factors for CNS metastasis development. Higher patient age and the location of the primary tumor in the upper or lower extremities were protective factors. In survival analysis, post-CNS metastasis, symptomatic disease, prior non-CNS metastases, CNS debut with multiple metastases, elevated LDH levels, and leptomeningeal involvement correlated with poorer survival. Conclusion: Predictive factors in the primary tumor independently associated with brain metastases include microscopic satellitosis, ulceration, higher Breslow index, and trunk location. Prognostic factors for lower survival in CNS disease include symptomatic disease, multiple CNS metastases, and previous metastases from different sites.

## 1. Introduction

Cutaneous melanoma represents 5% of all skin cancers, yet it causes 90% of skin cancer mortality [1]. Furthermore, melanoma is the third most common cancer that leads to brain metastases, after lung and breast cancer [2,3,4]. Further, the percentage of patients with metastatic melanoma who will develop brain metastases at some point in their disease has been calculated as between 40% and 60%, and central nervous system (CNS) metastases have been detected in up to 80% of autopsies of patients with metastatic melanoma [4,5,6,7].

Previous studies aimed to identify risk factors for CNS metastasis at the time of diagnosis. Notably, male sex, age (40–60), Breslow > 4 mm, ulceration, head and neck location, nodular subtype, and higher T stage were identified as key factors for the risk of developing brain metastases [8].

Historically, brain metastases have been associated with a poor prognosis, with median overall survival estimated to be approximately 4–6 months [7,8,9,10,11,12,13,14]. More recent studies have reported an improvement in median overall survival from 14 months to 23 months for metastatic melanoma since the introduction of targeted therapies and immune checkpoint inhibitors [15,16,17]. The CheckMate 204 trial reported that the 3-year overall survival rate was 71.9% for patients with asymptomatic brain metastases who received ipilimumab plus nivolumab [18].

The aim of our study was to describe risk factors associated with the development of brain metastases and to identify prognostic factors associated with survival once the patient develops brain metastasis.

## 2. Materials and Methods

A single-center cohort study was performed in the Melanoma Unit of the Hospital Clinic of Barcelona between January 1998 and September 2023. The data were collected from the electronic medical records of patients with a diagnosis of cutaneous melanoma and brain metastases developed at the time of diagnosis or during follow-up.

Patients with in situ melanoma and patients with brain metastasis who presented with a prior diagnosis of lung or breast cancer were excluded (Flowchart 1). To standardize AJCC in different time periods, a restaging of all primary melanomas was performed according to the AJCC 8th edition.

Since not all brain metastases could be confirmed histologically, diagnosis was performed by compatible magnetic resonance imaging or computed tomography, and in those cases where it was possible to perform a biopsy, the histopathologic report was used. Follow-up protocols were carried out at our center by a multidisciplinary tumor board of specialists in skin cancer and melanoma. The STROBE guidelines for the conducting and reporting of this observational study were followed.

### 2.1. Independent Variables

Collected data encompassed patient demographics and histological characteristics. CNS metastases characteristics were analyzed, covering number, size, bleeding, affected organs, symptoms, location, leptomeningeal involvement, and previous metastatic disease. Metastasis was categorized as deriving from 1, 2–5, or >5 lesions, and organ involvement as one, two-three, or >three. The largest metastasis size was reported as <10 mm, 10–25 mm, or >25 mm. Location was classified as supratentorial, infratentorial, multiple locations, or leptomeninges only. Additionally, leptomeningeal involvement alone was noted. The presence of metastases (debut or previous) and serum lactate dehydrogenase (LDH) concentration (≤ULN or >ULN) were recorded.

### 2.2. Dependent Variables

For the first predictive analysis, the presence of CNS metastases was coded as a dichotomous categorical variable.

In the post-metastasis survival analysis, the first dependent variable was melanoma-specific survival, calculated from the time of CNS melanoma metastases to the time of death from melanoma or the last follow-up visit. The second was overall survival, which was calculated from the time of CNS melanoma metastases to the time of death from any cause or the last follow-up visit.

### 2.3. Statistical Analysis

The distributions of categorical variables were reported as frequencies and percentages, and the distributions of continuous variables were reported as median and interquartile range (IQR). Pearson’s Chi-squared test was applied for categorical variables, and the trend test was applied for ordinal variables. The Wilcoxon rank sum test was used to compare continuous independent variables. The median follow-up of the cohort was calculated using the reverse Kaplan–Meier estimator applying the “Hist” functions in the ‘‘prodlim’’ package (v 2023.3.31) in R.

We utilized Sankey plots to represent the migratory patterns of AJCC stages in melanoma patients, leading to the development of brain metastasis. We also calculated the proportion of patients within each AJCC group who progressed. To describe the percentage of patients who will develop CNS metastases over time, we utilized the cumulative plot of survival.

To evaluate the association of covariables with brain metastasis development, we utilized multivariable logistic regression analysis.

After the development of brain metastasis, our objective was to assess the factors associated with poorer survival. Therefore, we performed univariate and multivariate analyses using Cox proportional hazard models for melanoma-specific survival. Models were fitted using the ‘‘coxph’’ function in the ‘‘survival’’ package (v 3.5.5) in R. Hazard ratio (HR) estimates were calculated to assess the effect of brain metastasis on melanoma-specific survival (MSS) adjusted for age at diagnosis of the brain metastasis, sex, primary tumor T score (AJCC 8th), presence of symptoms, levels of serum LDH, brain hemorrhage, number of brain metastasis, major diameter of metastasis, metastasis location, leptomeningeal involvement, and previous metastasis in other organs. We accounted for the changes in the therapeutic landscape by including a variable for treatment era in our survival analyses. Moreover, patients were grouped by treatment period: pre-2011 and post-2011, marking the introduction of targeted therapies and immune checkpoint inhibitors. This classification was included in the Cox proportional hazards model to adjust for the impact of different treatment regimens on survival outcomes.

Missing data were evaluated, classified as missing at random, and excluded from the analysis. To prevent immortal time bias in the setting of multiple primary melanomas, we used a worst-case analysis. To account for multiple comparisons, we applied the False Discovery Rate (FDR) method. This adjustment ensures that the likelihood of type I errors is minimized when performing multiple statistical tests.

All statistical analyses were performed using the computing environment R version 4.3.1 (16 June 2023) and RStudio (v 2023.6.1.524). A *p*-value of <0.05 was considered statistically significant.

## 3. Results

A total of 7010 patients with a histopathological diagnosis of melanoma were assessed for eligibility. After applying the inclusion and exclusion criteria, a total of 4718 patients were analyzed, of which 380 (8.05%) developed CNS melanoma metastases (Figure 1). The median follow-up time of the cohort using the inverse Kaplan–Meier estimator was 8.2 years (IQR 3.4–14.1). Clinicopathological characteristics of our study population at the time of initial diagnosis are summarized in Table 1.

The Sankey plot in Figure 2 illustrates the migration of patients who developed CNS metastases according to their initial AJCC 8th edition stage. The plot shows the percentage of patients from each stage who developed CNS metastases and those who did not. The highest proportion of CNS metastases originates from patients initially staged as IIID (53.8%) and IV (33.7%). Stages IIA and IIIA show similar percentages of CNS metastases, with approximately 7.7% and 7.3%, respectively. Stage IIC has a similar proportion to stage IIIB, with 15.1% and 18.5%, respectively. This visualization highlights the substantial risk of CNS metastases in later-stage melanoma patients. Additionally, it underscores that early-stage melanomas also contribute significantly in absolute numbers to the development of brain metastases.

In Figure 3, the cumulative plot of survival shows the percentage of patients who will develop CNS metastases over the years. At 5 years, the incidence rate in stages IIC was 17.7% (95% CI 10.5–24.4%), which was similar to stages IIIB with 20.6% (95% CI 13.6–27%) and IIIC with 22.7% (95% CI 17.9–27.3%). Furthermore, stage IIIA had an even lower incidence rate than stages IIA and IIB, at only 4.6% (95% CI 0.9–8.2%).

Multivariable logistic regression analysis to evaluate the association of covariables with brain metastasis development showed that CNS metastases decrease significantly in older patients. We noticed a reduced risk of metastasis development in the age group of 47.5–66 years and even lower in patients over 66 years. The odds ratios (OR) for these groups were 0.78 (95%CI 0.58–1.04; *p* < 0.001) and 0.47 (95%CI 0.35–0.64; *p* < 0.001), respectively, when compared to patients under 47.5 years of age. Moreover, a higher Breslow index correlated with an exponentially increased risk of developing CNS metastases with an OR of 2.7 (95%CI 1.9–4.1, *p* < 0.001) for T2, 3.7 (95%CI 2.4–5.6, *p* < 0.001) for T3, and 6.2 (95%CI 4.1–9.5, *p* < 0.001) when compared with T1. More importantly, the presence of ulceration and microscopic satellitosis was associated with a statistically significant elevation in this risk, with an OR of 2.4 (95%CI 1.8–3.2, *p* < 0.001) and 1.9 (95%CI 1.2–3.0, *p* < 0.007). Furthermore, when considering tumor location, primary tumors located on both the upper and lower extremities exhibited a decreased risk of CNS metastasis in comparison to those located on the trunk (Table 2).

Basal characteristics of the cohort at the time of CNS metastasis are summarized in Table 3. The median MSS of CNS metastatic patients was 5.73 months (95% Confidence Interval (CI): 5.06–6.53). For individuals experiencing CNS metastases at the onset of metastatic disease, the median time of survival was 7.46 months, while for patients whose metastases had spread to other organs before CNS, the median survival time was 5.39 months. Survival analysis using multivariate Cox regression models after the diagnosis of CNS metastasis showed that the presence of neurological symptoms is linked to higher specific mortality with an HR of 2.34 (95%CI 1.67–3.29; *p* < 0.001), as well as leptomeningeal involvement with an HR of 1.97 (95% IC 0.99–3.93; *p* < 0.054), however, it was not statistically significative. In addition, the presence of 2–5 lesions is associated with a worse MSS prognosis than a single metastatic lesion, with an HR of 1.53 (95% IC 1.05–2.22; *p* < 0.025) and more than 5 lesions had an HR of 2.45 (95% IC 1.65–3.65; *p* < 0.001), respectively, when compared with single lesions. Furthermore, another factor that influences survival negatively is an elevated level of LDH with an HR of 1.85 (95% IC 1.32–2.58; *p* < 0.001), while an indicator of better prognosis, with an HR of 0.66 (95% IC 0.44–0.99, *p* < 0.045), is the location of the metastasis as other than supratentorial (Table 4).

## 4. Discussion

Our study included 4718 melanoma patients, of whom 380 (8.05%) developed central nervous system metastases. We observed variations in CNS metastasis incidence rates across AJCC stages, with stage IIIC-D posing the highest risk. Moreover, when evaluating predictive risk factors for the development of brain metastasis, we observed that the risk of CNS metastases decreased with age, while an elevated Breslow index, ulceration, and microscopic satellitosis increased this risk, while tumors located on extremities rather than the trunk were associated with a lower risk. These findings provide new insights into the metastatic patterns of melanoma, which can be crucial for early detection and intervention.

Analyzing the baseline characteristics of the cohort, we observed that patients who developed central nervous system (CNS) metastases initially exhibited unfavorable tumor features: specifically, a median Breslow index exceeding a median of 3 mm, with over half presenting ulceration, aligning with findings from prior studies [2,19,20,21,22,23,24,25]. Notably, microsatellites were also seen in our study to be more prevalent in patients who subsequently developed CNS metastasis, occurring in 8.7% of cases vs. 2.3% in the control group. In a recent study by our research group, conducted by Riquelme-McLoughlin et al., we also identified microsatellites as an independent risk factor linked to poorer survival outcomes [26].

Examining metastasis occurrence across all AJCC substages, we find that stages IIIB through IV constitute about 50% of the CNS metastatic cohort. However, early stages, particularly stage IA, represent a significant portion of CNS progression cases (10.5%), attributed to their higher absolute numbers. Despite lower proportions, the substantial volume of individuals in early stages contributes significantly to metastatic cases. Moreover, risk analysis by AJCC groups aligns with expectations, revealing higher-risk stages with earlier metastasis development. Notably, AJCC Stage IIID stands out as the group at highest risk, with 53.8% developing CNS disease. Interestingly, many AJCC stages overlap; for example, stage IIIA behaves similarly to stage IIA, comprising 7.3% and 7.7% of total patients, respectively, showing that the AJCC is not a linear risk classification. Our results are in line with the study by Johannet et al., where patients with stage III melanoma were at a higher risk of developing brain metastases compared to those with stage II melanoma (21.4% vs. 14.0%, *p* = 0.002). However, individuals with stage IIC melanoma had a significantly higher rate of isolated first recurrences in the CNS compared to those with stage III disease (12.1% vs. 3.6%, *p* = 0.002). The risk of ever developing brain metastases was elevated for patients with stage IIC, IIIB, and IIIC disease, with the highest risk observed in patients with stage IIID disease (HR, 8.59; 95%CI: 4.11–17.97) [17].

Early identification of patients at high risk of CNS metastases is challenging. Notably, primary tumor factors such as microscopic satellitosis, ulceration, higher Breslow index, and trunk location serve as predictive indicators. Independent risk analysis shows younger age of onset correlates with higher risk, while those aged 66 or older are at a reduced risk. Breslow index, especially in tumors larger than 4 mm, amplifies risk six-fold, underscoring its prognostic significance. Davies et al. observed that the majority of patients who developed brain metastases had a primary tumor with an advanced Breslow index, with only 18% having an index < 1 mm [7]. Moreover, Zhang et al. and Gardner et al. identified male gender, ages between 40 and 60, Breslow thickness > 4 mm, ulceration, head and neck primary site, nodular subtype, and higher T stage as key risk factors associated with the development of brain metastases [8,27]. Interestingly, tumors situated in the upper and lower extremities exhibited a lower rate of progression to CNS compared to primary trunk tumors, suggesting a potential protective factor. This characteristic aligns with the findings described by Gardner et al. [27]. While some studies link the occurrence of brain metastases to male gender, in our study, male sex was not identified as an independent factor increasing the risk of CNS metastases [14,25,28,29]. Furthermore, the presence of microscopic satellitosis, a factor not previously reported, has emerged as an independent risk factor for the development of CNS disease, with an odds ratio of 1.90 (95%CI: 1.20 to 2.95). Our study becomes more significant after recent research by Riquelme-Mc Loughlin et al., which highlights microsatellites as an independent risk factor for reduced overall survival in melanoma [26].

In post-CNS metastasis survival analysis, factors contributing to a poorer prognosis include neurological symptoms and a higher lesion count, particularly when exceeding five, which is associated with diminished MSS. Elevated LDH levels were also linked to worse outcomes. However, the presence of hemorrhage and metastasis diameter did not significantly impact melanoma-specific survival. These findings align with a previous study by Glitza et al., where active disease outside the CNS, over three brain metastases, poor patient functional status, and leptomeningeal involvement were associated with poor survival [7]. Conversely, a more favorable prognosis was observed for metastases located other than supratentorial. Furthermore, in line with our findings, Vecchio et al. noted that asymptomatic patients exhibit significantly better survival [30]. On the other hand, other variables associated with the primary tumor were not associated with poor survival after central nervous system metastasis.

We believe that regular follow-up is essential for detecting metastasis in the early stages of relapse, particularly to identify asymptomatic patients, as our study identified a better outcome in individuals who exhibited no symptoms and presented with smaller tumor volumes. Looking ahead, it will be crucial to identify patients at risk of CNS relapse based on their basal characteristics and potentially through tumoral genetic tests such as gene expression profiling [31]. This could enable the selection of high-risk patients for a more intensive and targeted follow-up approach. The data from our study can be used to develop risk models for predicting CNS metastases, informing personalized follow-up strategies, and improving patient outcomes.

Our study’s retrospective nature introduces inherent heterogeneities in patient follow-up and treatment. Additionally, we were not able to analyze the molecular status of primary tumors due to the unavailability of comprehensive molecular panels, such as those for BRAF mutations, during the earlier years of our study period. Evaluating survival is challenging due to the diverse therapies applied to metastatic melanoma over the past decade. To address this, we grouped patients into two survival periods: pre-2011 and post-2011, marking the introduction of targeted therapies and immune checkpoint inhibitors. This classification helps to adjust for the impact of different treatment regimens on survival outcomes. Despite being a single-center study, our strength lies in analyzing a substantial cohort of 4718 patients, identifying 380 with CNS metastases.

## 5. Conclusions

In conclusion, our study identifies key risk factors for predicting CNS metastases in melanoma patients, with microsatellitosis being particularly significant. We also found that demographic and clinical factors such as advanced age, primary tumor thickness, and ulceration are associated with increased risk. Our analysis of prognostic factors influencing survival after CNS metastasis highlights the impact of neurological symptoms, the number of brain lesions, and elevated LDH levels on patient outcomes. These findings have important clinical implications for guiding neuroimaging follow-up strategies and tailoring treatment plans. By understanding both predictive and prognostic factors, healthcare providers can make more informed decisions in the management of melanoma, ultimately improving patient outcomes.

## Figures and Tables

**Figure 1 cancers-16-02272-f001:**
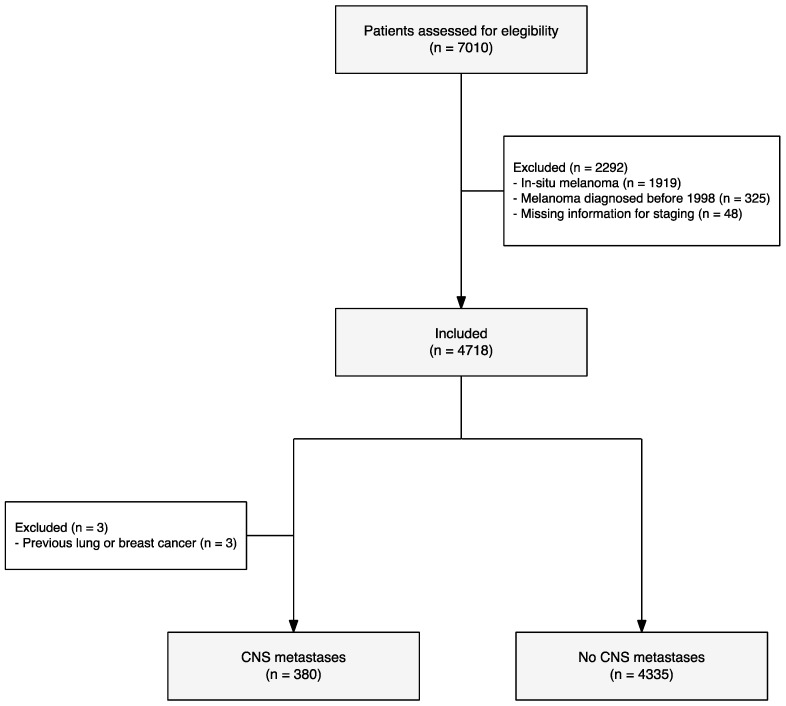
Flowchart of the cohort.

**Figure 2 cancers-16-02272-f002:**
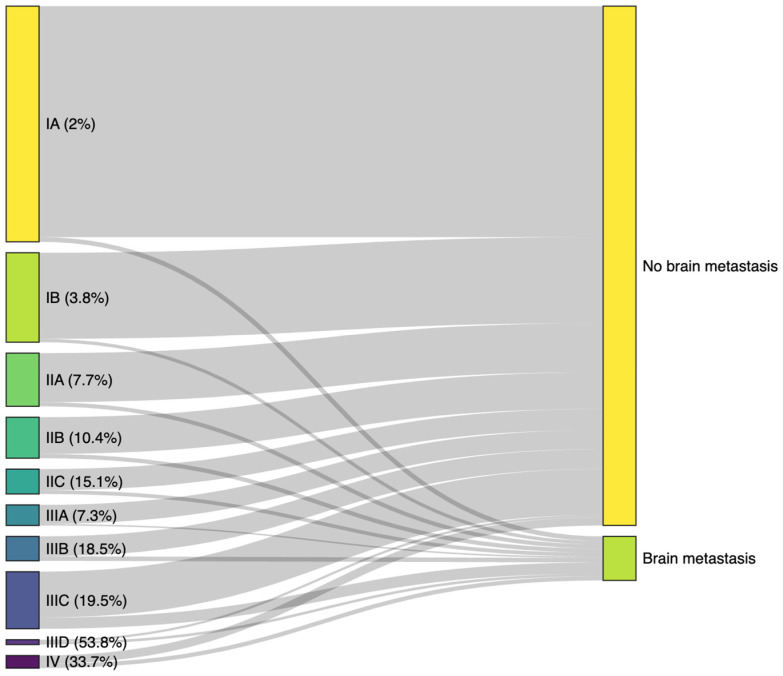
Sankey plot illustrating the migration of patients from initial AJCC 8th edition stages to CNS metastasis status. The width of each flow corresponds to the proportion of patients progressing to brain metastases or remaining metastasis-free. Higher stages, particularly IIID and IV, show a greater proportion of CNS metastases.

**Figure 3 cancers-16-02272-f003:**
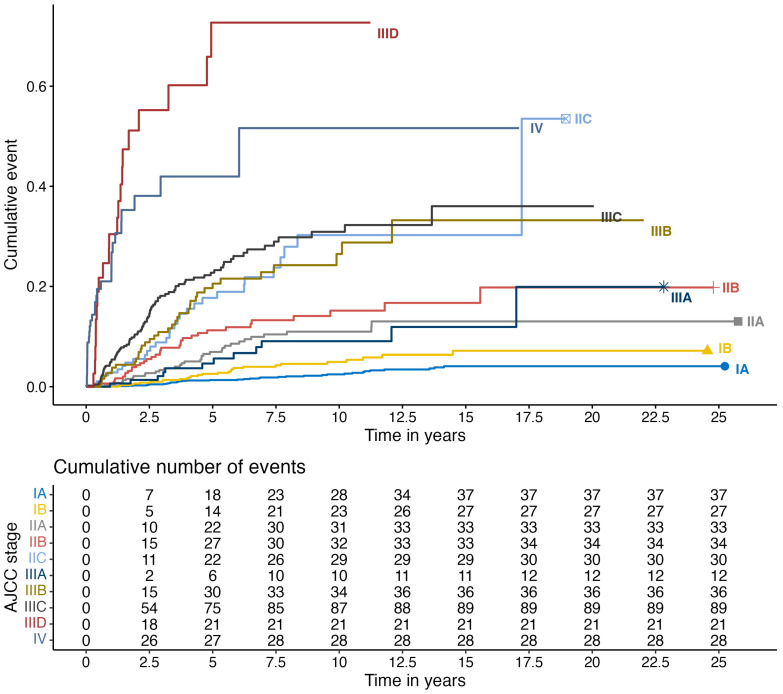
Cumulative plot for the incidence of central nervous system metastases.

**Table 1 cancers-16-02272-t001:** Basal characteristics of the cohort.

Characteristic	Overall, N = 4715	Stratified Groups		*p*-Value	q-Value
		No Brain Metastasis, N = 4335	Brain Metastasis, N = 380		
**Gender, n (%)**				<0.001	<0.001
Female	2410 (51.1%)	2249 (51.9%)	161 (42.4%)		
Male	2305 (48.9%)	2086 (48.1%)	219 (57.6%)		
**Age, Median (IQR)**	56.8 (42.8–70.3)	56.8 (42.9–70.6)	56.6 (41.3–67.8)	0.114	0.110
Missing values	9	9	0		
**8th version AJCC, n (%)**				-	-
IA	1889 (41.8%)	1852 (44.5%)	37 (10.5%)		
IB	717 (15.9%)	690 (16.6%)	27 (7.6%)		
IIA	426 (9.4%)	393 (9.4%)	33 (9.3%)		
IIB	328 (7.3%)	294 (7.1%)	34 (9.6%)		
IIC	199 (4.4%)	169 (4.1%)	30 (8.5%)		
IIIA	165 (3.7%)	153 (3.7%)	12 (3.4%)		
IIIB	195 (4.3%)	159 (3.8%)	36 (10.2%)		
IIIC	457 (10.1%)	368 (8.8%)	89 (25.2%)		
IIID	39 (0.9%)	18 (0.4%)	21 (5.9%)		
IV	101 (2.2%)	67 (1.6%)	34 (9.6%)		
Missing values	199	172	27		
**Melanoma of unknown primary, n (%)**	130 (2.8%)	91 (2.1%)	39 (10.3%)	<0.001	<0.001
Missing values	9	9	0		
**Breslow index (mm), Median (IQR)**	1.2 (0.7–2.7)	1.1 (0.6–2.5)	3.2 (1.5–6.0)	<0.001	<0.001
Missing values	123	84	39		
**Ulceration, n (%)**				<0.001	<0.001
Absent	3360 (75.7%)	3220 (78.3%)	140 (42.6%)		
Present	1081 (24.3%)	892 (21.7%)	189 (57.4%)		
Missing values	274	223	51		
**Mitotic index mm^2^, n (%)**				<0.001	<0.001
Zero Mitosis	1738 (44.7%)	1681 (46.7%)	57 (20.1%)		
One or more mitosis	2146 (55.3%)	1919 (53.3%)	227 (79.9%)		
Missing values	831	735	96		
**Histological subtype, n (%)**				<0.001	<0.001
Superficial spreading	2916 (65.5%)	2773 (67.1%)	143 (44.0%)		
Lentiginous malignant	289 (6.5%)	273 (6.6%)	16 (4.9%)		
Nodular	695 (15.6%)	582 (14.1%)	113 (34.8%)		
Other	299 (6.7%)	279 (6.8%)	20 (6.2%)		
Acral lentiginous	256 (5.7%)	223 (5.4%)	33 (10.2%)		
Missing values	260	205	55		
**Satellitosis, n (%)**				<0.001	<0.001
Absent	4581 (97.2%)	4234 (97.7%)	347 (91.3%)		
Present	134 (2.8%)	101 (2.3%)	33 (8.7%)		
**Regression, n (%)**				<0.001	<0.001
<50%	966 (31.5%)	915 (32.0%)	51 (24.6%)		
>50%	369 (12.0%)	361 (12.6%)	8 (3.9%)		
None	1734 (56.5%)	1586 (55.4%)	148 (71.5%)		
Missing values	1646	1473	173		
**Location, n (%)**				-	-
Trunk	2044 (44.9%)	1867 (44.4%)	177 (51.0%)		
Head and neck	644 (14.2%)	579 (13.8%)	65 (18.7%)		
Lower limbs	916 (20.1%)	879 (20.9%)	37 (10.7%)		
Upper limbs	546 (12.0%)	522 (12.4%)	24 (6.9%)		
Acral	333 (7.3%)	297 (7.1%)	36 (10.4%)		
Mucosa	64 (1.4%)	56 (1.3%)	8 (2.3%)		
Other	3 (0.1%)	3 (0.1%)	0 (0.0%)		
Missing values	165	132	33		

Legend Table 1: q-value: Adjusted *p*-value using the False Discovery Rate (FDR) method, which controls for the number of tests being performed to reduce false positives.

**Table 2 cancers-16-02272-t002:** Multiple Binomial Logistic Regression.

Characteristic	N	OR (95% CI) *^1^*	*p*-Value	q-Value *^2^*
**Age**	4401		<0.001	<0.001
* <47.5*		—		
* 47.5–66*		0.78 (0.58 to 1.04)		
* >66*		0.47 (0.35 to 0.64)		
**Gender**	4401		0.28	0.28
* female*		—		
* male*		1.14 (0.89 to 1.47)		
**T score**	4401		<0.001	<0.001
* pT1*		—		
* pT2*		2.77 (1.87 to 4.13)		
* pT3*		3.67 (2.42 to 5.60)		
* pT4*		6.19 (4.07 to 9.53)		
**Ulceration**	4401		<0.001	<0.001
* absent*		—		
* present*		2.41 (1.83 to 3.19)		
**Location**	4401		<0.001	<0.001
* trunk*		—		
* head and neck*		1.00 (0.72 to 1.39)		
* lower limbs*		0.45 (0.30 to 0.65)		
* upper limbs*		0.52 (0.32 to 0.82)		
* acral*		0.84 (0.55 to 1.26)		
* mucosa*		0.62 (0.24 to 1.37)		
**Satellitosis**	4401		0.007	0.009
* absent*		—		
* present*		1.90 (1.20 to 2.95)		

*^1^* OR = Odds Ratio, CI = Confidence Interval. *^2^* False discovery rate correction for multiple testing.

**Table 3 cancers-16-02272-t003:** Basal characteristics of the cohort at the time of metastasis.

Characteristic	N = 380
**Gender, n (%)**	
Female	161 (42.4%)
Male	219 (57.6%)
**Age at metastasis, Median (IQR)**	59.5 (46.5–70.6)
**Brain metastasis location, n (%)**	
Supratentorial	249 (71.3%)
Multiple locations	81 (23.2%)
Infratentorial	14 (4.0%)
Leptomeninges Exclusively	5 (1.4%)
Missing values	31
**Leptomeningeal involvement, n (%)**	
No leptomeningeal involvement	326 (93.4%)
Leptomeningeal metastasis	23 (6.6%)
Missing values	31
**Brain hemorrage, n (%)**	
Absent	256 (72.3%)
Present	98 (27.7%)
Missing values	26
**Previous metastasis, n (%)**	
Debut	82 (22.5%)
Previous metastasis	282 (77.5%)
Missing values	16
**Number of other organs affected, n (%)**	
None	77 (21.4%)
1	103 (28.7%)
2–3	114 (31.8%)
>3	65 (18.1%)
Missing values	21
**Symptoms, n (%)**	
Asymptomatic	169 (47.2%)
Symptomatic	189 (52.8%)
Missing values	22
**Diametre of largest brain metastasis, n (%)**	
<10 mm	103 (33.9%)
10–25 mm	113 (37.2%)
>25 mm	88 (28.9%)
Missing values	76
**Lactate dehydrogenase level (LDH), n (%)**	
≤ULN	186 (61.6%)
>ULN	116 (38.4%)
Missing values	78

**Table 4 cancers-16-02272-t004:** Univariate and multivariate Cox regression tables for melanoma-specific and overall survival.

			Melanoma-Specific Survival		Overall Survival	
Variable	Values	N (%)	HR (Univariable)	HR (Multivariable)	HR (Univariable)	HR (Multivariable)
**Sex**	female					
	male	161 (42.8)	1.00 (0.80–1.24, *p* = 0.978)	0.96 (0.71–1.30, *p* = 0.782)	1.00 (0.81–1.25, *p* = 0.967)	0.96 (0.71–1.29, *p* = 0.773)
**Age**	<47.5	215 (57.2)				
	47.5–66	130 (34.6)	1.27 (0.98–1.64, *p* = 0.069)	1.08 (0.76–1.55, *p* = 0.658)	1.28 (0.99–1.65, *p* = 0.061)	1.08 (0.75–1.54, *p* = 0.684)
	>66	135 (35.9)	1.31 (0.99–1.72, *p* = 0.055)	1.18 (0.81–1.72, *p* = 0.391)	1.30 (0.99–1.71, *p* = 0.058)	1.14 (0.78–1.66, *p* = 0.487)
**T score**	pT1	111 (29.5)				
	pT2	44 (13.1)	0.80 (0.54–1.17, *p* = 0.249)	1.01 (0.61–1.68, *p* = 0.967)	0.79 (0.54–1.17, *p* = 0.246)	1.00 (0.60–1.66, *p* = 0.993)
	pT3	75 (22.3)	0.81 (0.55–1.17, *p* = 0.262)	0.91 (0.55–1.51, *p* = 0.717)	0.84 (0.58–1.21, *p* = 0.346)	0.94 (0.57–1.56, *p* = 0.818)
	pT4	87 (25.8)	0.82 (0.58–1.17, *p* = 0.276)	0.93 (0.58–1.51, *p* = 0.778)	0.84 (0.59–1.19, *p* = 0.317)	0.93 (0.57–1.50, *p* = 0.751)
**Symptoms**	Asymptomatic	131 (38.9)				
	Symptomatic	169 (47.5)	2.33 (1.86–2.92, *p* < 0.001)	2.34 (1.67–3.29, *p* < 0.001)	2.31 (1.85–2.90, *p* < 0.001)	2.31 (1.65–3.25, *p* < 0.001)
**LDH levels**	≤ULN	187 (52.5)				
	>ULN	186 (61.6)	1.94 (1.51–2.48, *p* < 0.001)	1.85 (1.32–2.58, *p* < 0.001)	1.92 (1.50–2.46, *p* < 0.001)	1.82 (1.30–2.54, *p* < 0.001)
**Brain hemorrhage**	absent	116 (38.4)				
	present	256 (72.7)	1.44 (1.13–1.83, *p* = 0.003)	1.25 (0.86–1.82, *p* = 0.242)	1.42 (1.11–1.80, *p* = 0.005)	1.25 (0.86–1.81, *p* = 0.249)
**Number of brain metastasis**	1 lesion	96 (27.3)				
	2–5 lesions	121 (35.3)	2.07 (1.57–2.72, *p* < 0.001)	1.53 (1.05–2.22, *p* = 0.025)	2.09 (1.59–2.75, *p* < 0.001)	1.55 (1.07–2.25, *p* = 0.020)
	>5 lesions	125 (36.4)	2.38 (1.77–3.20, *p* < 0.001)	2.45 (1.65–3.65, *p* < 0.001)	2.43 (1.81–3.26, *p* < 0.001)	2.44 (1.64–3.63, *p* < 0.001)
**Diameter of largest brain metastasis**	<10 mm	97 (28.3)				
	10–25 mm	103 (34.0)	1.52 (1.14–2.04, *p* = 0.005)	0.99 (0.69–1.43, *p* = 0.953)	1.50 (1.13–2.01, *p* = 0.005)	0.97 (0.67–1.39, *p* = 0.861)
	>25 mm	112 (37.0)	1.80 (1.32–2.45, *p* < 0.001)	0.91 (0.59–1.42, *p* = 0.686)	1.76 (1.30–2.39, *p* < 0.001)	0.89 (0.57–1.39, *p* = 0.611)
**Location**	Supratentorial	88 (29.0)				
	Other location	248 (71.5)	1.21 (0.94–1.55, *p* = 0.140)	0.66 (0.44–0.99, *p* = 0.045)	1.20 (0.94–1.54, *p* = 0.139)	0.66 (0.44–0.98, *p* = 0.041)
**Leptomeningeal involvement**	No leptomeningeal involvement	99 (28.5)				
	Leptomeningeal metastasis	324 (93.4)	1.94 (1.27–2.97, *p* = 0.002)	1.97 (0.99–3.93, *p* = 0.054)	1.92 (1.25–2.93, *p* = 0.003)	1.90 (0.96–3.79, *p* = 0.067)
**Previous metastasis**	Debut	23 (6.6)				
	Previous metastasis	79 (21.9)	1.60 (1.22–2.10, *p* = 0.001)	1.87 (1.26–2.78, *p* = 0.002)	1.63 (1.24–2.14, *p* < 0.001)	1.92 (1.30–2.86, *p* = 0.001)
**Year of diagnosis**	<2011	281 (78.1)				
	≥2011	114 (30.3)	0.74 (0.59–0.93, *p* = 0.010)	0.71 (0.52–0.98, *p* = 0.040)	0.73 (0.58–0.92, *p* = 0.008)	0.70 (0.51–0.97, *p* = 0.031)

## Data Availability

The data presented in this study are available in this article.

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
