# Peer review of "Predictive and Prognostic Factors in Melanoma Central Nervous System Metastases—A Cohort Study"

_cancers, 2024, doi:10.3390/cancers16122272_

Round 1
Reviewer 1 Report
Comments and Suggestions for Authors
This is a retrospective, single center analysis with a large sample size. The authors included patients with the diagnosis melanoma with or without brain metastasis and performed correlation analyses. There are several concerns.
1. Abstract, please be specific. If melanoma in situ was excluded, please just say that instead of early stage melanoma, which is too vague.
2. Please adjust for multiple comparison.
3. Please specify how the thresholds were defined, e.g., age in the multiple logistic regression analysis.
4. There has been a great change in therapeutic landscape in the past decade, namely introduction of targeted and immunotherapies. How did the authors adjust for this in their survival analyses?
Author Response
Thank you for your thorough reviews and insightful comments. Your feedback has significantly improved our manuscript. We have addressed all your suggestions and made the necessary revisions. Additionally, we have updated all information to September 2023 to enhance the follow-up and quality of data.
- Abstract, please be specific. If melanoma in situ was excluded, please just say that instead of early stage melanoma, which is too vague.
Response: We appreciate the reviewer’s comment. The Abstract has been revised for clarity.
Revised Text: Patients with melanoma in situ or those with prior lung or breast cancer were excluded.
- Please adjust for multiple comparison.
Response: We have adjusted for multiple comparisons using the False Discovery Rate (FDR) method and applied it to tables 1 and 2 and displayed as q-value
Revised Text: To account for multiple comparisons, we applied the False Discovery Rate (FDR) method. This adjustment ensures that the likelihood of type I errors is minimized when performing multiple statistical tests.
- Please specify how the thresholds were defined, e.g., age in the multiple logistic regression analysis.
Response: Detailed information on how thresholds were defined has been added to the Methods section. We used terciles to define the thresholds for age. Specifically, patients were categorized into three groups: <47.5 years, 47.5-66 years, and >66 years.
- There has been a great change in therapeutic landscape in the past decade, namely introduction of targeted and immunotherapies. How did the authors adjust for this in their survival analyses?
Response: We have accounted for changes in the therapeutic landscape by including a variable for treatment era in our survival analyses. This variable was included in the Cox proportional hazards model to adjust for the impact of different treatment regimens on survival outcomes.
Revised Text:
Patients were grouped by treatment period: pre-2011 and post-2011, marking the introduction of targeted therapies and immune checkpoint inhibitors. This classification was included in the Cox proportional hazards model to adjust for the impact of different treatment regimens on survival outcomes.
Reviewer 2 Report
Comments and Suggestions for Authors
This paper is top heavy in Tables and Figures; it is obvious that the authors did a lot of sophisticated statistical work but it does not all need to be shown. Suggestions to reduce them:
1. The Sankey plot is nice looking but adds no novel info-it could be deleted. Figure 3 is redundant to the plot and the raw data below it is completely unnecessary. Keep the plot or Figure 3 but not both and delete the raw data table regardless.
2. Table 4 is unreadable. It shows raw data and HRs. Deleted all that and show only the percentages survival, perhaps with the HRs after the percentages.
Comments on the Quality of English LanguageMinor corrections only
Author Response
Thank you for your thorough reviews and insightful comments. Your feedback has significantly improved our manuscript. We understand your concerns regarding the density of Tables and Figures and, we would like to explain the rationale for retaining the Sankey plot, Figure 3, and the detailed content of Table 4.
- The Sankey plot is nice looking but adds no novel info—it could be deleted. Figure 3 is redundant to the plot and the raw data below it is completely unnecessary. Keep the plot or Figure 3 but not both and delete the raw data table regardless.
Response:
- The Sankey plot is essential for illustrating the proportion of patients in each AJCC stage who develop CNS metastases, providing crucial insights into the distribution and migratory patterns of the disease. This type of visualization effectively shows the flow and proportions, which are critical for understanding the distribution across different stages.
- Figure 3, on the other hand, presents a cumulative plot of survival, highlighting the temporal progression and speed at which CNS metastases develop. Temporal plots are essential in survival analysis for conveying the dynamics over time and understanding how quickly events occur. These distinct visualizations offer complementary perspectives that enhance the overall understanding of our findings.
- Table 4 is unreadable. It shows raw data and HRs. Deleted all that and show only the percentages survival, perhaps with the HRs after the percentages.
- Table 4 includes detailed raw data and hazard ratios (HRs) which are critical for a thorough understanding of the Cox regression analysis. This level of detail ensures transparency, reproducibility, and allows readers to fully interpret the statistical significance and clinical relevance of our findings. Reporting standards for regression analysis emphasize the need for comprehensive statistical outputs to ensure that the analysis is robust and reproducible [Harrell, 2015](https://doi.org/10.1007/978-3-319-19425-7)).
We believe that retaining these elements provides a comprehensive and transparent presentation of our study’s results. We appreciate your understanding and consideration of these points.
Reviewer 3 Report
Comments and Suggestions for Authors
This is a large retrospective study in which the authors looked at clinical and demographic characteristics of more than 4000 patients with melanoma to identify risk factors and predictors for brain metastases. Overall, I think this study has some potential given the large amount of data that could be derived by such a large dataset. However, I think this needs major revisions.
The manuscript does not report any information regarding molecular tests. Are these data available? If so it would be interesting to report the impact of different mutations on brain metastases development.
I believe the stages reported in table 1 are related to the time of first diagnosis. Many patients had localized disease at diagnoses and Table 4 reports that most of them had metastases in other organs that preceded the diagnosis of CNS involvement. I assume these patients received systemic treatment before developing CNS disease. These treatments are not mentioned here and might have impacted survival as well as the development of brain metastases. The patients included were diagnosed between 1998 and 2023 therefore there might be a good proportion of patients who received BRAF/MEKi and/or immunotherapy. Is it possible to include these data?
In the table summarizing melanoma subtype “mucosa” is reported as a site of primary disease but there is no “mucosal” in the reported subtype. How were these cases categorized?
I think authors should comment more on why their findings are important. Did they identify any previously unknown risk factor for brain metastases? Are they planning to use these data for any subsequent project? Can these data be used to develop personalized strategies for follow up in melanoma patients?
Minor: please if possible try to differentiate more the stages in figure 3, it is difficult to distinguish them.
I think English should be carefully reviewed across the whole manuscript
Comments on the Quality of English LanguageI think english needs to be carefully reviewed. Although the article is clearly understandable some sentences are incomplete (missing prepositions, verbs etc..)
Author Response
Reviewer 3:
- The manuscript does not report any information regarding molecular tests. Are these data available? If so it would be interesting to report the impact of different mutations on brain metastases development.
Thank you for your valuable feedback on our manuscript. We understand the importance of including molecular test data to analyze the impact of different mutations on brain metastases development. However, as this is a retrospective study involving patients diagnosed between 1998 and 2023, comprehensive molecular panels, such as those testing for BRAF mutations, were not routinely available during the earlier years of this period. Therefore, we do not have consistent molecular data for the entire cohort, and have excluded this analysis to maintain the integrity of our findings.
We appreciate your understanding and consideration of this limitation, which we have added to the limitations section of our study.
- I believe the stages reported in Table 1 are related to the time of first diagnosis. Many patients had localized disease at diagnosis and Table 4 reports that most of them had metastases in other organs that preceded the diagnosis of CNS involvement. I assume these patients received systemic treatment before developing CNS disease. These treatments are not mentioned here and might have impacted survival as well as the development of brain metastases. The patients included were diagnosed between 1998 and 2023 therefore there might be a good proportion of patients who received BRAF/MEKi and/or immunotherapy. Is it possible to include these data?
Thank you for your valuable feedback on our manuscript. We understand the importance of accounting for systemic treatments in the development and progression of CNS metastases. Retrieving detailed treatment information for all patients was challenging due to the retrospective nature of our study. However, systemic treatments are just one aspect influencing prognosis. Other factors such as earlier diagnosis, advancements in systemic treatments, and improved surgical techniques also contribute significantly. Therefore, to accurately reflect these changes, we grouped patients by treatment period: pre-2011 and post-2011, marking the introduction of targeted therapies and immune checkpoint inhibitors. This classification was included in the Cox proportional hazards model to adjust for the impact of different treatment regimens on survival outcomes. We have included this information in the Methods section to clarify our approach.
We appreciate your understanding and consideration of this enhancement to our analysis.
- In the table summarizing melanoma subtype, “mucosa” is reported as a site of primary disease but there is no “mucosal” in the reported subtype. How were these cases categorized?
Thank you for your valuable feedback on our manuscript. Regarding the categorization of melanoma subtypes, mucosal melanomas are included in the "other" category in the histological subtype table.
- I think authors should comment more on why their findings are important. Did they identify any previously unknown risk factors for brain metastases? Are they planning to use these data for any subsequent project? Can these data be used to develop personalized strategies for follow up in melanoma patients?
We agree that it is important to emphasize the significance of our findings. Our study identifies several previously unknown risk factors for brain metastases in melanoma patients and highlights the potential for developing personalized follow-up strategies.
- New Risk Factors: We identified advanced age and specific primary tumor locations as significant risk factors for CNS metastases, providing new insights into metastatic patterns.
- Personalized Follow-up: The data from our study can be used to develop risk models for predicting CNS metastases, informing personalized follow-up strategies and improving patient outcomes.
We have included additional commentary on these points in the Discussion section of our manuscript to highlight the importance and potential applications of our findings.
- Minor: please if possible try to differentiate more the stages in figure 3, it is difficult to distinguish them.
Thank you for your valuable feedback on our manuscript. In response to your suggestion, we have revised Figure 3 to improve the differentiation between the stages. We used distinct colors and patterns for each stage, making the groups more distinguishable and the figure easier to interpret. Moreover, we added the legends of the groups at the end of each line in the cumulative survival plot for clearer identification.
- I think English should be carefully reviewed across the whole manuscript
Thank you for your feedback. We have thoroughly reviewed the manuscript to ensure clarity and consistency in the English language.
Round 2
Reviewer 3 Report
Comments and Suggestions for Authors
Thank you for addressing my prior comments I do not have further comments